# A Salvaging Strategy Enables Stable Metabolite Provisioning among Free-Living Bacteria

Sebastian Gude,[a] Gordon J. Pherribo,[a] Michiko E. Taga[a]

aDepartment of Plant & Microbial Biology, University of California, Berkeley, Berkeley, California, USA

**ABSTRACT**   All organisms rely on complex metabolites such as amino acids, nucleotides, and cofactors for essential metabolic processes. Some microbes synthesize these fundamental ingredients of life *de novo*, while others rely on uptake to fulfill their metabolic needs. Although certain metabolic processes are inherently "leaky," the mechanisms enabling stable metabolite provisioning among microbes in the absence of a host remain largely unclear. In particular, how can metabolite provisioning among free-living bacteria be maintained under the evolutionary pressure to economize resources? Salvaging, the process of "recycling and reusing," can be a metabolically efficient route to obtain access to required resources. Here, we show experimentally how precursor salvaging in engineered *Escherichia coli* populations can lead to stable, long-term metabolite provisioning. We find that salvaged cobamides (vitamin $B_{12}$ and related enzyme cofactors) are readily made available to nonproducing population members, yet salvagers are strongly protected from overexploitation. We also describe a previously unnoted benefit of precursor salvaging, namely, the removal of the nonfunctional, proliferation-inhibiting precursor. As long as compatible precursors are present, any microbe possessing the terminal steps of a biosynthetic process can, in principle, forgo *de novo* biosynthesis in favor of salvaging. Consequently, precursor salvaging likely represents a potent, yet overlooked, alternative to *de novo* biosynthesis for the acquisition and provisioning of metabolites in free-living bacterial populations.

**IMPORTANCE**   Recycling gives new life to old things. Bacteria have the ability to recycle and reuse complex molecules they encounter in their environment to fulfill their basic metabolic needs in a resource-efficient way. By studying the salvaging (recycling and reusing) of vitamin $B_{12}$ precursors, we found that metabolite salvaging can benefit others and provide stability to a bacterial community at the same time. Salvagers of vitamin $B_{12}$ precursors freely share the result of their labor yet cannot be outcompeted by freeloaders, likely because salvagers retain preferential access to the salvaging products. Thus, salvaging may represent an effective, yet overlooked, mechanism of acquiring and provisioning nutrients in microbial populations.

**KEYWORDS**   salvaging, metabolite provisioning, partial metabolite privatization, intracellular metabolites, cobamides, cobalamin, vitamin $B_{12}$, metabolic interactions, free-living bacteria, microbial ecology

The general mechanisms enabling stable, long-term metabolite provisioning among free-living bacteria remain poorly understood (1–3). While host-associated bacteria may count on a consistent supply of metabolites from their host, their free-living counterparts, particularly in unstructured environments, may not. Not only do the latter have no reliable access to metabolites provided by a host, but they also lack simple means for positive assortment (4), i.e., the ability of metabolite providers to preferentially supply metabolites to population members that benefit the provider, which is typically enabled by structured environments. In the absence of positive assortment, nonproducing population members may easily invade, overexploit, and displace metabolite providers (5–8),

Address correspondence to Michiko E. Taga, taga@berkeley.edu.

The authors declare no conflict of interest.

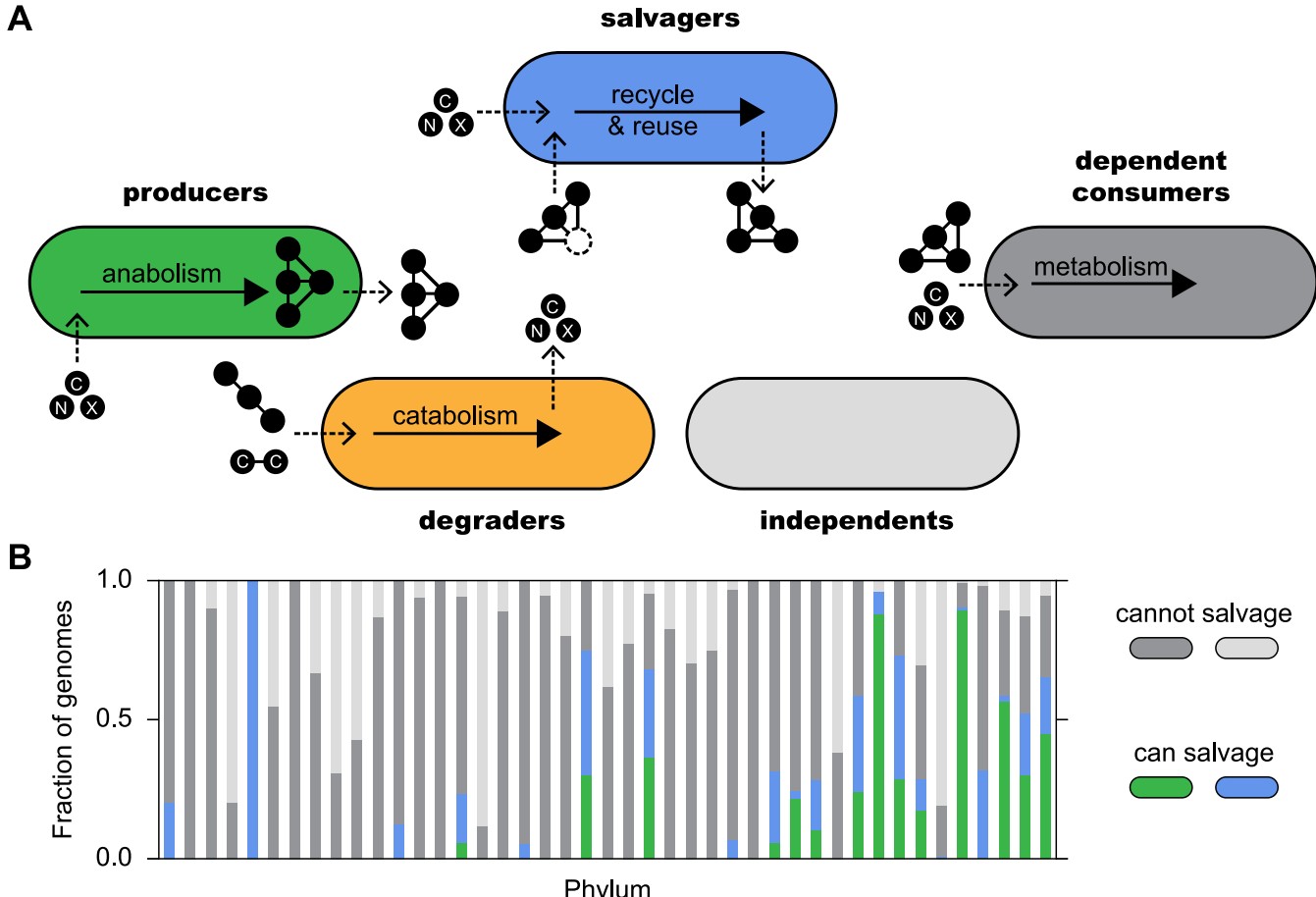

**FIG 1** Metabolic archetypes in bacterial populations. (A) Illustration of distinct metabolic archetypes in bacterial populations. Producers create complex metabolites (black circles connected by lines) *de novo* from simple precursors (black circles) such as carbon- and nitrogen-containing compounds. Synthesized complex metabolites may be released and may benefit others. Dependent consumers take up simple compounds and complex metabolites to fulfill their metabolic needs. Dependence on complex metabolites may be facultative or obligate. Dependent consumers do not release metabolites. Degraders break down complex compounds such as polysaccharides or polypeptides. Degradation products are often accessible to others. Salvagers take up compounds from the environment and complete metabolite biosynthesis. Salvaged compounds may be released to the benefit of others. Independents neither create nor utilize a given metabolite. The metabolic archetype of an organism may vary for different metabolites; e.g., a producer of one metabolite may be a consumer, degrader, salvager, or even be independent of another metabolite. C, carbon source; N, nitrogen source; X, generic compound. (B) Distribution of predicted cobamide archetypes among sequenced bacteria (30). The colors are as defined for panel A. The phyla are listed in Table S1.

causing an irreversible reduction in community diversity and metabolic capacity. The risk of overexploitation is not limited to fully public goods but can even occur when providers possess the ability to retain exclusive access to some portion of the metabolite pool, as this private pool alone may be insufficient to support their long-term proliferation.

Bacteria require a diverse set of metabolites. These essential ingredients of life can be obtained through *de novo* biosynthesis or via uptake from the environment (9). Metabolite provisioning among bacteria is frequently enabled by an intricate network of metabolic interactions (10). Members of complex microbial communities can be broadly categorized according to their metabolic archetype for a particular metabolite as producers, degraders, salvagers, dependent consumers, and independents (Fig. 1A; this study focuses exclusively on the interplay between salvagers and dependent consumers). While independents, by definition, do not partake in interactions involving the metabolite, dependent consumers (i.e., nonproducing population members) critically rely on metabolically active producers, degraders, or salvagers to fulfill their metabolic needs. Since certain biological functions are inherently "leaky" (11–14), some metabolites, such as degradation products of extracellularly processed polypeptides and polysaccharides (15, 16) or by-products of overflow metabolism (17), are readily supplied as "public goods." However, this type of metabolite provisioning is

inherently limited to particular classes of metabolites, such as externally processed polymeric compounds.

An alternative mode of gaining access to metabolites is salvaging, the process of "recycling and reusing" (18). Bacterial salvaging is most commonly associated with iron acquisition (19–21). In aerobic environments, many microbes release high-affinity iron-chelating siderophores to gain access to otherwise insoluble $Fe^{3+}$, yet bacterial salvaging is by no means limited to the uptake and reuse of inorganic substances such as metals. Bacteria can also take up a variety of environmental metabolites in lieu of synthesizing them *de novo*. Classes of metabolites known to be salvaged include sugars, amino acids, nucleotides, and vitamins (22–27). Notably, metabolite salvaging is not restricted to complete, terminal metabolites. Chemically stable precursors, for example intermediates of central metabolism or compounds involved in specialized metabolism, can be salvaged as well (28, 29).

While the genetic and biochemical basis of salvaging is well understood for many metabolites due to in-depth studies in a few bacterial species, less is known about the ecological ramifications of salvaging in multimember bacterial populations (22–26, 28). Here, as an example, we focus on the salvaging of the cobamide precursor, cobinamide (Cbi), and explore how the salvaging of metabolite precursors can shape bacterial population dynamics and stability. Cobamides (vitamin $B_{12}$ and related enzyme cofactors) are cobalt-containing cofactors used in diverse metabolic pathways, and cobamide precursor salvaging is widely predicted among sequenced bacterial genomes (30) (Fig. 1B; Table S1). In this process, the incomplete and nonfunctional cobamide precursor Cbi is taken up and, in the presence of the required gene products for cobamide precursor salvaging, combined with a nucleoside containing a "lower ligand" base of variable structure to form a complete, functional cobamide. By studying cobamide precursor salvaging, we demonstrate that salvaging can be an effective strategy enabling stable metabolite provisioning. Salvagers readily release assembled cobamides into the environment, where they can be utilized by nonproducing population members, but we found that salvagers were strongly protected from overexploitation. These findings indicate that precursor salvaging may be a highly effective, generalizable way of enabling metabolite provisioning, even when means for positive assortment are absent.

## RESULTS

**Salvaged metabolites are readily provisioned.** To experimentally investigate precursor salvaging in the context of mixed free-living bacterial populations, we generated cobamide-dependent *Escherichia coli* populations (Fig. 2A; see Materials and Methods for details). Salvagers (Sal) proliferated in glycerol minimal medium only when either the complete cobamide vitamin $B_{12}$ ($B_{12}$) or the precursor Cbi and the lower ligand, 5,6-dimethylbenzimidazole (DMB), which together can be salvaged and assembled into functional $B_{12}$ inside the cell, were supplied (Fig. 2B, blue symbols). As expected, no proliferation was observed in the presence of Cbi and DMB for dependent consumers (Dep), which have a disrupted cobamide precursor salvaging pathway (Fig. 2B, gray symbols).

Mixed salvager-dependent populations proliferated in the presence of Cbi and DMB (Fig. 2B). While salvagers were expected to proliferate, it remained unclear whether dependents would also be able to proliferate given that they rely on external provisioning of the cobamide $B_{12}$. Previous studies reported a need to engineer or evolve "overproduction" strains to enable provisioning of intracellular metabolites such as amino acids (11, 31). However, conversely, a recent report demonstrated the general potency of vitamins and cofactors to promote stable metabolic interactions (32). To dissect individual contributions to the overall increase in population size of the mixed salvager-dependent population, we established a calibrated fluorescence-based assay to track the dynamics of each subpopulation in well stirred culture conditions over time (Fig. S1). The increase in the total population size was found to be driven by proliferation of both salvagers and dependents,

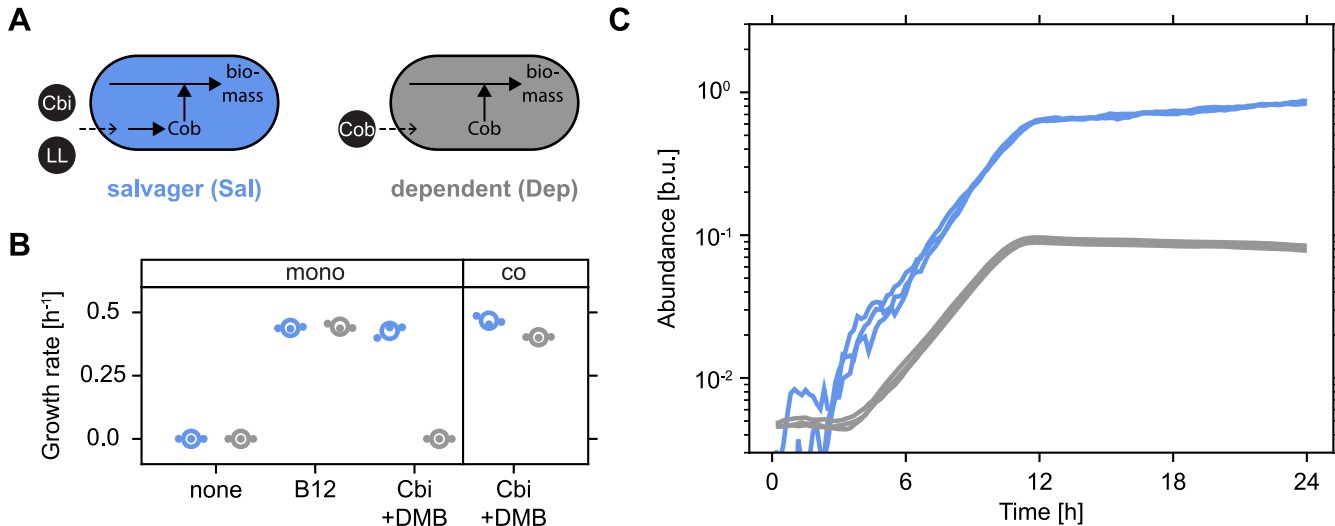

**FIG 2** Salvaged cobamides are readily provisioned. (A) Illustration of cobamide archetypes. (B) Exponential growth rates of salvagers (Sal, blue) and dependents (Dep, gray) in glycerol minimal medium with various additions (10 nM each). The strains were cultured alone (mono) or together (co, initial ratio: 1:1). Global means are shown as open circles, and biological replicates are shown as dots. The technical precision of growth rate estimates was approximately ±0.025 $h^{-1}$ (see Materials and Methods for details) (N = 3). (C) Proliferation of cocultured salvagers (blue) and dependents (gray). The initial ratio was 1:1 (N = 3). Abbreviations: Cbi, incomplete, nonfunctional cobamide precursor cobinamide; LL, lower ligand, a required component of a cobamide; Cob, complete, functional cobamide; B12, vitamin $B_{12}$; DMB, 5,6-dimethylbenzimidazole, the lower ligand of $B_{12}$; b.u., bacterial units (see Materials and Methods for details).

indicating that the salvaged cobamides were readily released into the environment as a public good and made available (i.e., provisioned) to the dependents (Fig. 2B and C).

Details of the proliferation dynamics differed between the two subpopulations. Salvagers proliferated with no lag and experienced a growth rate of approximately 0.47 $h^{-1}$ (Fig. 2B and C), comparable to their performance when cultured alone (Fig. 2B; technical precision of growth rate estimation: approximately ±0.025 $h^{-1}$; see Materials and Methods for details). Notably, dependents experienced an extended lag before proliferation commenced at a slightly reduced growth rate of approximately 0.40 $h^{-1}$ (see Materials and Methods for technical limitation in growth rate estimation), and their final abundance, which they attained at approximately the same time as salvagers ceased to proliferate, was only about one-tenth of the level achieved by salvagers (Fig. 2C). These findings suggested that the benefits of the salvaged cobamides were asymmetrically accessible to salvagers and dependents, even though mechanisms for positive assortment (4), such as spatial exclusion of dependents in structured environments (6), were inaccessible under these (well stirred) culture conditions.

**Salvaging "preference" determines population dynamics.** We next explored the potential limits of the unprompted cobamide release and provisioning observed in mixed salvager-dependent populations. We reasoned that salvaging-based proliferation and metabolite provisioning should depend on the quantity and chemical identity of the available precursors. To test this hypothesis, we systematically varied external lower ligand supplementation and monitored proliferation of cocultured salvagers and dependents. Proliferation dose-response curves in the presence of Cbi and various levels of DMB indicated that salvagers successfully proliferated at all tested concentrations of the externally supplied lower ligand (Fig. 3A). Conversely, proliferation of dependents ceased completely below 100 pM of externally supplied DMB, again pointing toward a preferential access of salvagers to the assembled cobamide. A similar trend was observed when a chemically different lower ligand, 2-methyladenine (2MA; lower ligand of the cobamide factor A [FA]), was supplied (Fig. 3B). Again, the proliferation of only the dependents ceased at low levels of external lower ligand supplementation, while salvagers continued to proliferate. Dependent proliferation was observed only at a level of 2MA approximately 2 orders of magnitude higher compared to DMB, indicating that chemical identity indeed played an important role. The difference in $EC_{50}$ of the growth

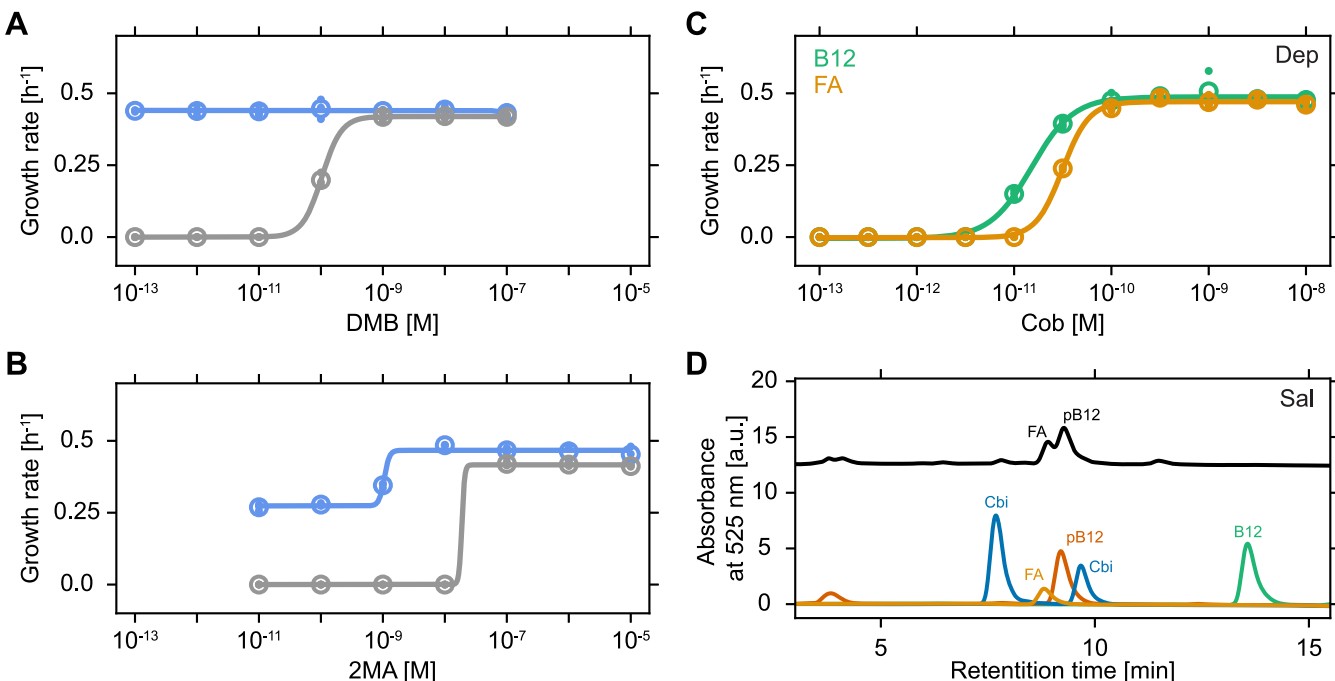

**FIG 3** Salvaging "preference" shapes population dynamics. (A, B) Growth rates of Sal (blue) and Dep (gray) when cocultured at an initial ratio of 1:1 in glycerol minimal medium supplemented with the cobamide precursor Cbi (10 nM) and various concentrations of external lower ligand, DMB (A), and 2-methyladenine (2MA) (B). Observed differences in salvager growth rates at low levels of lower ligand are likely caused by experiment-to-experiment variations in the amount of *de novo* synthesized lower ligand, as salvager growth rates at the lowest measured level of lower ligand supplementation were consistent with their growth rates in the absence of lower ligand (0.44 $h^{-1}$ in panel A and 0.27 $h^{-1}$ in panel B). (C) Growth rate of Dep in glycerol minimal medium with various concentrations of the cobamides $B_{12}$ (green, $EC_{50} = 1.5 \times 10^{-11} \pm 1.8 \times 10^{-12}$ M) and FA (yellow, $EC_{50} = 3.1 \times 10^{-11} \pm 2.0 \times 10^{-12}$ M). $EC_{50}$ is the cobamide concentration at half-maximal growth rate; 95% confidence intervals are given. In panels A to C, global means are shown as open circles, biological replicates are dots, and sigmoidal fits are lines (N = 3). (D) High-performance liquid chromatography (HPLC) analysis of cell extracts of Sal cultured in glycerol minimal medium supplemented with Cbi (10 nM) in the absence of an externally supplied lower ligand (black). Standards of cobinamide (Cbi, blue), factor A (FA, yellow), pseudo-$B_{12}$ (pB12, red), and vitamin $B_{12}$ (B12, green) are shown. Abbreviations: a.u., arbitrary units. The technical precision of growth rate estimates was approximately $\pm0.025$ $h^{-1}$ (see Materials and Methods for details).

rate (concentration at half-maximal growth rate) of Dep between the two media conditions, as estimated by fitting to four-parameter dose-response equations, was not statistically significant (two-tailed *t* test, $P > 0.99$), likely due to the absence of data points in the transition region ($10^{-8}$ M > 2MA < $10^{-7}$ M) of Fig. 3B. We reasoned that this difference in dependent proliferation between the two chemically distinct lower ligands could be caused by a difference in the metabolic "preference" for the respective cobamides, which has been reported for several bacterial species (33–35). Alternatively, a difference in the efficiency of the precursor salvaging pathway for installation of the two lower ligands could be decisive. Proliferation dose-response curves of dependents cultured alone showed that a lower concentration of B12 compared to FA was required for dependent proliferation, yet this difference in metabolic preference between the two cobamides was relatively minor (approximately 2-fold; two-tailed *t* test, $P < 10^{-16}$) (Fig. 3C). These findings indicate that the drastic difference in dependent proliferation between the supplementation with the chemically distinct lower ligands DMB and 2MA was not due to a metabolic preference for the cobamides $B_{12}$ and FA but was instead mainly caused by a difference in lower ligand preference of the precursor salvaging pathway. Similar observations of a salvaging preference in bacteria have previously been reported (36, 37).

The proliferation dose-response curves of cocultured salvagers and dependents contained an additional noteworthy detail (Fig. 3A and B). Namely, proliferation of salvagers was observed for levels of externally supplied lower ligand that, even if conversion to complete cobamide was 100% efficient, were below the minimal level of externally supplied cobamide required for proliferation (Fig. 3C). Thus, salvagers were able to generate proliferation-supporting levels of cobamides in the absence of sufficient external lower

ligand supplementation, presumably by incorporating internally produced lower ligands. Using high-performance liquid chromatography (HPLC) on cell extracts obtained from salvagers cultured in the presence of Cbi with no externally supplied lower ligand, we identified these "default" cobamides as a mixture of FA (lower ligand: 2MA) and pseudo-$B_{12}$ ($pB_{12}$, lower ligand: adenine) based on comparison of their respective retention times and characteristic peak spectra with purified standards (Fig. 3D; Fig. S4). Thus, as observed previously (38), salvagers can supplement limiting external precursor supply by tapping into their internal metabolite pool to obtain *de novo* synthesized lower ligands (here 2MA and adenine) to generate required cobamides.

**Salvaging can eliminate negative effects of metabolite accumulation.** Precursor salvaging seemed not to incur a noticeable metabolic cost under our culture conditions, as salvagers proliferated at comparable growth rates (i.e., identical within the technical precision of approximately $\pm$ 0.025 $h^{-1}$ of our measurements) in the presence of either the cobamide $B_{12}$ or the precursors Cbi and DMB (Fig. 2B). Nevertheless, the fact that metabolically valuable salvaged cobamides were readily released and provisioned to dependents remained puzzling, particularly because it invites overexploitation by dependents, which may cause a collapse of the entire mixed population (5–8). In an attempt to gain some basic understanding of the factors that may facilitate the unprompted release of the metabolically valuable salvaged cobamides, we tried taking a more abstract perspective on the precursor salvaging process by comparing its functioning to other similarly structured cellular activities. On the most basic level, cobamide precursor salvagers take up metabolites, modify them intracellularly, and finally release them back into the extracellular environment. While the unprompted release of metabolically valuable salvaged metabolites begs an explanation, the similarly structured process of antibiotic inactivation certainly does not. Here, toxic compounds are taken up, internally converted into less harmful forms, and subsequently released into the environment (39). We wondered whether in addition to its "constructive" characteristic, i.e., the generation of complete, functional cobamides, cobamide precursor salvaging could also have benefits through its "consuming" characteristic, i.e., the assimilation of the incomplete, nonfunctional cobamide precursor Cbi. The latter could be beneficial if the presence of high levels of metabolically valuable precursors interferes with proliferation, for example, if structural similarities between Cbi and B12 interfere with cellular functions via competitive binding to enzymes. Accumulation to proliferation-inhibiting levels has been noted for various metabolites (17, 40), and their consumption has been reported to stabilize commensal interactions (41).

Proliferation dose-response curves of dependents cultured alone on various levels of the cobamide $B_{12}$ in the absence and presence of high levels (10-fold higher than the highest tested level of $B_{12}$) of Cbi indeed revealed a clear proliferation-inhibiting effect (Fig. 4A). The presence of high levels of environmental Cbi not only increased the $EC_{50}$ of the proliferation dose-response curve by about 5-fold (Fig. 4A, black arrow; two-tailed $t$ test, $P < 10^{-6}$) but also induced the appearance of a considerable lag before proliferation was initiated for the lowest $B_{12}$ levels that supported proliferation in the presence of surplus Cbi (Fig. 4B). Thus, although Cbi is metabolically beneficial as a precursor for salvaging, it clearly has negative effects when present in excess.

We reasoned that this proliferation-inhibition caused by surplus Cbi could potentially be overcome, or at least be diminished, by the consuming character of precursor salvaging. Indeed, salvagers cultured under the same conditions as described above did not exhibit the lag previously observed for dependents (Fig. S2; Fig. 4B), yet the inhibitory effect of Cbi seen at higher $B_{12}$ levels remained unchanged (Fig. 4C, black arrow). Cobamide precursor salvaging in the absence of external lower ligand supplementation is likely limited by the internal availability of *de novo* synthesized lower ligands, which, under our conditions, could be insufficient to assimilate sufficient amounts of the proliferation-inhibiting Cbi. As a direct test, we supplied salvagers with a 10-fold excess of externally supplied lower ligand. Under these conditions, salvagers were indeed able to completely relieve the detrimental effects of surplus Cbi (Fig. 4D). Consequently, the consuming character of precursor salvaging, i.e., the uptake and

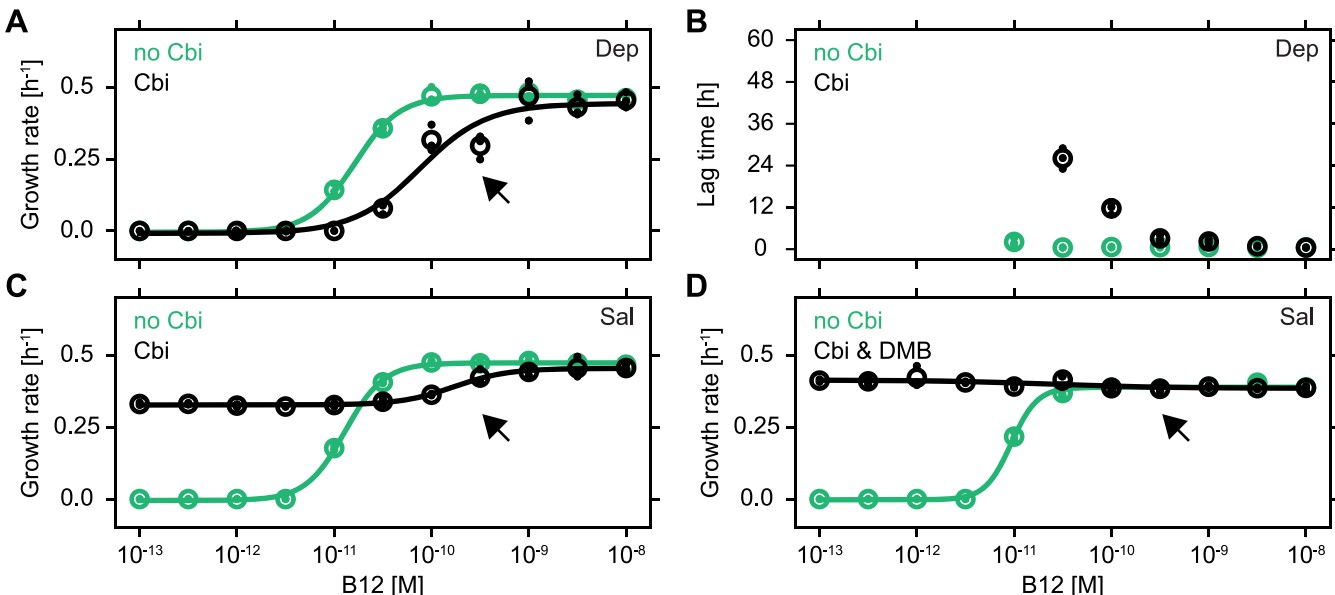

**FIG 4** Salvaging can relieve proliferation-inhibition. (A to C) Growth rate (A) and lag time (B) of Dep and growth rate of Sal (C) in glycerol minimal medium as a function of the concentration of $B_{12}$ in the absence (green) and presence (black) of the cobamide precursor Cbi ($10^{-7}$ M). Global means are shown as open circles, biological replicates are dots, and sigmoidal fits are lines (N = 3). (D) Same conditions as C, but with the lower ligand DMB [$10^{-6}$ M] also added (black) (N = 3). The technical precision of growth rate estimates was approximately $\pm 0.025$ h$^{-1}$ (see Materials and Methods for details).

assimilation of proliferation-inhibiting precursors and the subsequent export of complete cobamides, can be beneficial to salvagers, and thus may (partially) explain the unprompted release of salvaged metabolites.

To explore whether these Janus-like (i.e., two-sided) benefits of precursor salvaging were reflected in its regulatory logic, we constructed a fluorescent reporter for the expression of the major genes involved in the cobamide precursor salvaging process (i.e., cobUST) to determine whether salvaging is regulated by precursor abundance. While gene expression changed slightly (approximately 4-fold) over time, we did not observe differential gene expression between conditions that required precursor salvaging (supplementation of Cbi and DMB) and conditions that did not (supplementation of $B_{12}$) (Fig. S3). Thus, our results suggested that cobamide precursor salvaging in *E. coli* was not regulated at the level of gene expression under our culturing conditions and that the precursor salvaging pathway was expressed independently of the need for cobamide precursor salvaging.

**Salvagers persist stably in coculture.** Unprompted metabolite release in the absence of mechanisms for positive assortment bears the risk of overexploitation, potentially causing an associated collapse of a metabolically interconnected population if nonproducing members (i.e., dependents) possess a proliferation advantage over their productive counterparts (i.e., salvagers) (42). In particular, faster-proliferating dependents could cause an arrest in salvager proliferation by quickly depleting the publicly available cobamide and carbon if salvagers provisioned too lavishly, making themselves temporarily dispensable. Loss of salvagers would inevitably result in a collapse of the entire metabolically interconnected population once externally available cobamide is depleted. Alternatively, salvagers could be able to persist independently of the dependents' proliferation properties by securing an adequate access to cobamides prior to releasing them into the extracellular environment. As an initial step to assess the risk of population collapse in mixed salvager-dependent populations, we created faster-proliferating mutant variants by moving the salvager and dependent genotypes into a closely related *E. coli* MG1655 GB-1 mutant background (Sal$^{GB-1}$ and Dep$^{GB-1}$) (43). The GB-1 background contains only two mutations (in the glycerol kinase *glpK* and the RNA polymerase subunit $\beta'$ *rpoC*, https://www.biocyc.org/gene?orgid=ECOLI&id=RPOC-MONOMER) relative to wild-type *E. coli* MG1655 and confers a growth rate advantage of approximately 45% compared to the original salvager and dependent in glycerol minimal

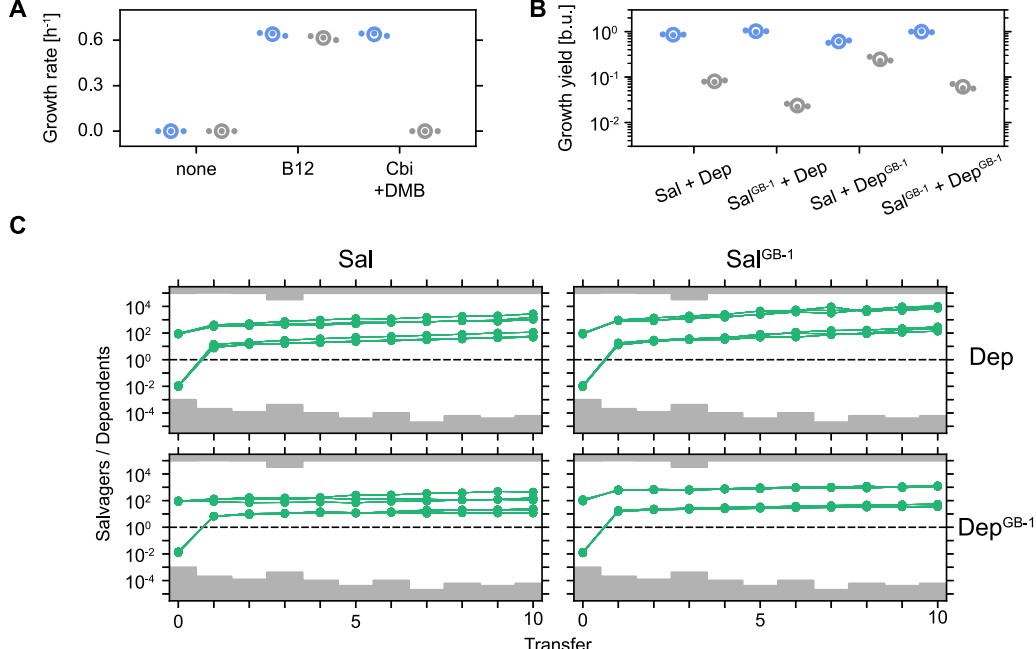

**FIG 5** Salvagers are protected from overexploitation. (A) Growth rates of Sal<sup>GB-1</sup> (blue) and Dep<sup>GB-1</sup> (gray) when cultured alone in glycerol minimal medium with various additions (10 nM each). Technical precision of growth rate estimates: approximately $\pm$ 0.025 h$^{-1}$ (see Materials and Methods for details). (B) Growth yields after 24 h of cocultured salvagers (Sal or Sal<sup>GB-1</sup>, blue) and dependents (Dep or Dep<sup>GB-1</sup>, gray) in glycerol minimal medium supplemented with Cbi (10 nM) and DMB (10 nM). Initial ratio: 1:1. In panels A and B, global means are shown as open circles, and biological replicates are dots (N = 3). (C) Population ratio (green dots) as function of transfer for cocultured salvagers (Sal or Sal<sup>GB-1</sup>) and dependents (Dep or Dep<sup>GB-1</sup>) undergoing daily 1:100 (vol/vol) proliferation-dilution cycles in glycerol minimal medium supplemented with Cbi (10 nM) and DMB (10 nM). Solid green lines connect individual cultures through subsequent transfers. Gray areas mark transfer-specific estimates of population ratio detection limits (for details see Materials and Methods) (N = 3).

medium (compare Fig. 2B and 5A) (43). Combinatorial culturing of all four possible salvager-dependent pairs as 1:1 mixtures in the presence of excess levels of Cbi and DMB resulted in expected changes in the final population composition (Fig. 5B). Specifically, when cocultured with Sal, Dep reached only about one-tenth of the abundance of Sal after 24 h, in agreement with our previous observations (Fig. 2C). Cocultured Sal<sup>GB-1</sup> and Dep<sup>GB-1</sup> showed a similar pattern. Dep proliferation was reduced when it was cultured with the faster-proliferating Sal<sup>GB-1</sup> mutant, whereas the contrary effect was observed if the dependent (Dep<sup>GB-1</sup>) had a proliferation advantage over the salvager (Sal). Together, these findings demonstrated how relative changes in the maximally achievable growth rate can reshape the composition of metabolically linked populations. Moreover, it suggested that dependents may indeed have the ability to overexploit salvagers by quickly depleting the public cobamide pool if they have sufficiently large growth rate advantages.

To explore the consequences of growth rate differences on long-term population dynamics and stability more rigorously, we exposed all four mixed salvager-dependent populations to a sequence of proliferation-dilution cycles (Fig. 5C). Mixed populations were initialized with either salvagers or dependents 100-fold in the majority, with a fixed total initial population size. The culture medium, including glycerol, Cbi, and DMB, was replenished at each transfer. Interestingly, salvagers were able to persist, and even achieved clear numerical dominance, in all four mixed populations (Fig. 5C). These findings demonstrated that salvagers were strongly protected from overexploitation even if dependents were able to proliferate faster by as much as 45%. Details of long-term population dynamics varied among the four mixed populations. Differences in maximally achievable growth rate was a dominant factor determining the rate at which population ratios changed over time. For example, faster-proliferating salvagers

outcompeted slower-proliferating dependents at a higher rate compared to slower-proliferating salvagers (Fig. 5C, Sal$^{GB-1}$/Dep versus Sal/Dep; two-tailed $t$ test for difference in slopes of $\log_{10}$(population ratio) from transfer 1 onwards; slopes of Sal$^{GB-1}$/Dep were greater than of Sal/Dep in all 18 cases, with 10 of 18 statistically significant; see Materials and Methods for details). Similarly, faster-proliferating dependents fared better than their slower-proliferating counterparts, as they were typically outcompeted at a lower rate (Fig. 5C, Sal$^{GB-1}$/Dep$^{GB-1}$ versus Sal$^{GB-1}$/Dep and Sal/Dep$^{GB-1}$ versus Sal/Dep; two-tailed $t$ test for difference in slopes of $\log_{10}$(population ratio) from transfer 1 onwards; slopes Sal$^{GB-1}$/Dep$^{GB-1}$ were smaller than of Sal$^{GB-1}$/Dep in all 18 cases, with 17 of 18 statistically significant; slopes of Sal/Dep$^{GB-1}$ were smaller than of Sal/Dep in 16 of 18 cases, with 12 of 16 statistically significant; and the remaining 2 slopes of Sal/Dep$^{GB-1}$ were larger than of Sal/Dep, with 0 of 2 statistically significant). Interestingly, the population ratio of Sal/Dep changed at a slightly higher rate than that of Sal$^{GB-1}$/Dep$^{GB-1}$ (Fig. 5C, Sal/Dep versus Sal$^{GB-1}$/Dep$^{GB-1}$; two-tailed $t$ test for difference in slopes of $\log_{10}$(population ratio) from transfer 1 onwards; slopes of Sal/Dep were larger than of Sal$^{GB-1}$/Dep$^{GB-1}$ in all 18 cases, with 16 of 18 statistically significant), although we were unable to detect differences in maximally achievable growth rate within each of the strain pairs within the technical precision of our measurements (Fig. 2B and 5A).

## DISCUSSION

Cobamide precursor salvaging has been experimentally observed and is predicted to be widespread among bacteria (30, 44). Here, we showed that (i) salvaging of cobamide precursors does not incur a drastic metabolic burden, (ii) salvaged cobamides are readily released and provisioned to others, (iii) cobamide precursor salvaging provides benefits through the generation of complete functional cofactors, as well as through the removal of nonfunctional, proliferation-inhibiting precursors, and (iv) organisms engaging in cobamide precursor salvaging are intrinsically protected from overexploitation, likely by retaining preferential access to the salvaged cobamides prior to release (i.e., partial metabolite privatization).

While not all of these features may be transferable to salvaging of other types of metabolites or in all bacteria, some certainly will. For example, removal of proliferation-inhibiting metabolites may be widespread as high levels of many compounds, such as cysteine (40), can lead to disturbances of optimal metabolic fluxes and thus hinder proliferation. Furthermore, (partial) privatization, which has also recently been proposed in the context of other microbial activities (7, 45, 46), is likely applicable to a broad set of metabolites, including amino acids, nucleotides, and cofactors, that are inherently processed inside the cell. Thus, organisms generating these intracellular metabolites, either via salvaging or *de novo* biosynthesis, may be naturally protected from overexploitation by nonproducing population members, and therefore may facilitate the emergence and stable maintenance of metabolite provisioning interactions.

The unprompted release of metabolically valuable salvaged cobamides initially seemed puzzling, as salvagers could potentially reduce their metabolic burden by precisely tuning their uptake and salvaging activity to their current needs while completely privatizing all salvaged cobamides. The absence of complete privatization of salvaged cobamides could be related to the dual benefits of the salvaging process. In particular, a certain degree of continuous cobamide release may have evolved as a compromise to accommodate both the constructive (i.e., generation of complete, functional cobamides) and the consuming (i.e., the assimilation of the incomplete, nonfunctional cobamide precursor Cbi) aspects of cobamide salvaging.

Our findings indicate how intracellular metabolic processes that can innately enable (partial) retention of metabolites inside the cell are fundamentally distinct from extracellular processes such as degradation of polysaccharides (15). In the latter case, metabolites immediately become public goods that have to be taken up from the environment via finite affinity import systems (47, 48), thus inherently limiting the ability to privatize resources. In contrast, intracellular metabolic processes, such as cobamide precursor

salvaging, may sequester a portion of the salvaged cobamides inside the cell for private use prior to releasing them into the extracellular environment, thus ensuring preferential access via partial metabolite privatization. The observed extended lag time of dependents when cocultured with salvagers (Fig. 2C) is consistent with such a mode of release. However, more insights into cobamide release mechanisms, such as the identification and characterization of the molecular basis of cobamide export systems in bacteria, which remain unknown to date, are required to gain a deeper understanding of the factors shaping these metabolic interactions. Ultimately, our findings demonstrate the ability of intracellular metabolic processes to establish stable metabolic interactions, with important ramifications for community stability and the protection of metabolite-producing population members.

The differences in the maximally achievable growth rate were a dominant factor determining coculture behavior (Fig. 5B and C). One exception to this trend was the observed difference in the rate of change in population ratio of Sal/Dep and $Sal^{GB-1}/Dep^{GB-1}$ (Fig. 5C). This apparent discrepancy may simply be explained by growth rate differences that were below the precision limit of our assays. Alternatively, the observed difference in long-term competition dynamics could also be caused by phenotypic variations other than differences in exponential growth rate in the strain backgrounds. For example, differences in cobamide release during stationary phase could shift the long-term population dynamics to the benefit of the dependents. The faster-proliferating GB-1 mutant was originally obtained by laboratory evolution of *E. coli* maintained in exponential phase (43). Detailed multiomics characterization of the GB-1 mutant indeed indicated that its proteome shows a clear reduction in stationary phase and stress-induced proteins (49). This difference may cause a reduction in survival in stationary phase, leading to release of intracellular material by dying cells, which may enable increased access of the dependents to metabolites that are normally sequestered.

Salvaging, the concept of recycling and reusing, provides a resource-efficient way to use environmental metabolites and inorganic substances that were released due to metabolite overflow or cell lysis. It is important to note that metabolite salvaging is not limited to the uptake and completion of precursors, which was the focus of this study. Salvaging can also be combined with partial catabolism and metabolite modification to interconvert complete yet enzymatically unfavorable metabolites into more useful forms (50, 51). Taken together, salvagers of various metabolites likely play an important, underappreciated role in many natural bacterial populations by acting as hubs facilitating efficient community functioning by rerouting and interconverting environmental metabolic fluxes. For example, precursor salvaging has been reported for thiamine (52, 53), and the highly interconnected nature of amino acid biosynthesis provides ample opportunity for the provisioning of stable precursors (54). Innate protection from overexploitation by nonproducing population members positions salvaging of intracellular metabolites as an effective mechanism to act in a broad range of environmental conditions, even those in which positive assortment is not accessible.

## MATERIALS AND METHODS

**Strains and plasmids.** All strains were derivatives of *E. coli* strain MG1655. $Sal^{GB-1}$ and $Dep^{GB-1}$ were derived from *E. coli* MG1655 mutant GB-1 (43). Strain $Sal^{GB-1}$ was constructed by introducing into strain GB-1 the *metE*::$Kan^R$ allele from donor strain JW3805-1 (55) via P1 transduction and subsequently removing the $Kan^R$ marker by transient introduction of the plasmid pCP20 encoding the FLP recombinase (56). To generate strain $Dep^{GB-1}$, the *cobUST* operon was deleted via lambda red recombineering (56), and the gene *cobC* was replaced via P1 transduction as described above. After each step, the selective marker was removed. Gene deletions were verified via PCR and phenotyping of growth in glycerol minimal medium supplemented with either methionine, vitamin $B_{12}$, cobinamide and DMB, or no addition.

Fluorescent marker and reporter plasmids are based on pETMini (51) and were assembled via isothermal cloning (57). Insert sequences were verified via Sanger sequencing. All bacterial strains and plasmids are listed in Tables S2 and S3.

**Media and culturing.** For proliferation and serial passaging assays, 2 mL glycerol minimal medium (50 mM $KPO_4$, 67 mM NaCl, 7.6 mM $(NH_4)_2SO_4$, 500 $\mu$M $MgSO_4$, 1.25 $\mu$M $Fe_2(SO_4)_3$, 0.2% [vol/vol] glycerol, pH 7.4) supplemented with 1 g/liter methionine was inoculated with single colonies from LB plates (10 g/liter tryptone, 5 g/liter yeast extract, 5 g/liter NaCl, 15 g/liter Bacto agar) and incubated at 37°C with aeration (200 rpm) until saturation was reached after approximately 24 h. Cultures were transferred into

2 mL fresh glycerol minimal medium supplemented with methionine as 1:100 (vol/vol) dilutions, incubated at 37°C with aeration, and harvested in midexponential phase (approximate $OD_{600}$ of 0.3 to 0.6). The cells were washed three times by centrifugation at 14,000 rpm followed by resuspension in unsupplemented minimal medium. Optical density ($OD_{600}$) was adjusted to $10^{-1}$, and if required, the cultures were mixed 1:1 (vol/vol) or, as indicated, to obtain mixed populations. If necessary, media were supplemented with 25 mg/liter kanamycin to retain plasmids.

For proliferation assays, precultured and OD-adjusted cultures were transferred as 1:10 (vol/vol) dilutions into glass-bottom 96-well culture plates (CellVis) containing a final volume of 200 $\mu$L fresh glycerol minimal medium supplemented with either 10 nM vitamin $B_{12}$, 10 nM cobinamide and 10 nM 5,6-dimethylbenzimidazole (DMB), or no additions, if not indicated otherwise. The 96-well culture plates were sealed with an evaporation seal (Breathe-Easy sealing membrane) and incubated at 37°C with continuous shaking in a multiwell plate reader (Tecan Spark) for 36 or 60 h. Absorbance at 600 nm and fluorescence (cyan fluorescent protein [CFP]: excitation at 455/5 nm, emission at 475/5 nm; yellow fluorescent protein [YFP]: excitation at 514/5 nm, emission at 550/30 nm) were recorded every 10 min. Fluorescence measurements were converted into units equivalent to $OD_{600}$ (see below and Fig. S1). Fluorescence was recorded only for experiments containing two strains. All proliferation experiments were performed in biological triplicates started on the same day. Proliferation phenotypes showed little day-to-day variation.

For serial passaging assays, precultured and OD-adjusted mixed cultures were transferred as 1:10 (vol/vol) dilutions into clear plastic 96-well culture plates (Corning) containing a final volume of 200 $\mu$L fresh glycerol minimal medium supplemented with 10 nM cobinamide and 10 nM DMB. Wells without cultures were filled with 200 $\mu$L distilled $H_2O$ ($dH_2O$). The 96-well culture plates were sealed with an evaporation seal (AeraSeal, EXCEL Scientific) and incubated at 37°C while shaking at 1200 rpm in a heated plate shaker (Southwest Science). The cultures were transferred every 24 h as 1:100 (vol/vol) dilutions into fresh glycerol minimal medium supplemented with 10 nM cobinamide and 10 nM DMB. At each transfer, the samples were taken for composition analysis at the Flow Cytometry Facility in the Cancer Research Laboratory at UC Berkeley. Prior to flow cytometry analysis (BD LSRFortessa cell analyzer), the samples were diluted 1:1,000 (vol/vol) into unsupplemented minimal medium. Forward scatter (FSC; $V = 350$, log), side scatter (SSC; $V = 280$, log), FITC ($V = 650$, log), and AmCyan ($V = 750$, log) were recorded. Event detection was triggered by a threshold in SSC of 500. The samples were run at approximately 300 to 600 events/s, $10^5$ events were collected per sample, and lines were flushed with 10% bleach, rinse solution, and $dH_2O$ between sample runs. The serial transfer experiment was performed in biological triplicates started on the same day.

For corrinoid extractions, 2 mL glycerol minimal medium supplemented with 100 mg/liter methionine was inoculated with single colonies of strains grown on LB plates and incubated at 37°C with aeration until saturation was reached. The cells were washed twice by centrifugation at 14,000 rpm and resuspension in unsupplemented minimal medium, diluted 1:500 (vol/vol) into 500 mL glycerol minimal medium supplemented with 10 nM cobinamide, incubated at 37°C with aeration for 24 h, and harvested for metabolite extraction, purification, and profiling (see below).

**Evaluation of proliferation dynamics.** For proliferation assays, population dynamics were evaluated in Python (58) (version 3.8) with the following steps: (i) Fluorescence-to-absorbance conversion (if applicable): Fluorescence-to-absorbance conversion factors were estimated in a day-, strain-, and medium-dependent manner for each fluorescent channel. Conversion factors were obtained as replicate-averaged slopes of linear fits (*NumPy* [59], version 1.21.2, function *polyfit*) of raw fluorescence versus raw absorbance ($A_{600}$) in the range $A_{600} > 0$ and $A_{600} < 0.3$ of cultures containing only a single marker plasmid. Raw fluorescence reads were then converted into equivalent units of raw absorbance by multiplication of the conversion factors with the raw fluorescence reads. (ii) Absorbance-to-abundance conversion: All reads were pathlength-corrected to obtain abundances in arbitrary bacterial units (b.u.; 1 b.u. is equivalent to an $OD_{600}$ of 1). (iii) Blank subtraction: For reads based on absorbance measurements, blanks were estimated in a well-specific manner as the difference of the mean of the first two time points and the known initial abundance. For reads based on fluorescence measurements, blanks were estimated in a global manner as means over all wells containing cultures that did not carry the specific fluorescent marker plasmid. (iv) Smoothing and shortening: Blank-corrected abundance time series were smoothed with a centered rolling mean filter (version 1.3.2 *pandas DataFrame* function *rolling*) using a window width of seven data points with all data points equally weighted and a minimal window width of one point. Thereafter, smoothed abundance time series were shortened to 24 or 60 h. (v) Extraction of proliferation parameters: Growth rate estimates were obtained via a Theil-Sen estimator (*SciPy* [60], version 1.7.1, module *stats.mstats* function *theilslopes*), which was applied to log-transformed abundances. Estimation was restricted to data points for which the abundance was larger than $1.5 \times 10^{-2}$ b.u. and smaller than $6 \times 10^{-2}$ b.u. (Fig. 3C and 4A, C, and D) or for which the abundance was larger than $1.5 \times 10^{-2}$ b.u. and the latest data point for which the rolling-window growth rate estimate (see below) was larger than 0.1 $h^{-1}$ (Fig. 2B; 3A and B; and 5A). The growth rate was set to zero if fewer than seven data points fell into the estimation interval. Rolling-window growth rate estimates were obtained by applying a Theil-Sen estimator (*SciPy* [60], version 1.7.1, module *stats.mstats* function *theilslopes*) to log-transformed abundances in a centered rolling-window fashion (*pandas*, version 1.3.2, *DataFrame* function *rolling*) using a window width of 19 data points with all data points equally weighted and a minimal window width of 10 data points. Lag time estimates were obtained by linear interpolation of growth rates and intercepts obtained from the Theil-Sen estimator to known log-transformed initial abundances. Lag time was set to negative infinity for nongrowing populations. (vi) Extraction of dose-response curve parameters (if applicable): Dose-response curve parameters were obtained via nonlinear fitting

(*SciPy* [60], version 1.7.1, module *optimize* function *curve_fit*) of extracted means of proliferation parameters to a four-parameter dose-response equation:

$$y = y_{\text{bottom}} + \quad x^n \times \frac{y_{\text{top}} - y_{\text{bottom}}}{x^n + EC50^n}$$

with variables $x$ (effector concentration) and $y$ (proliferation parameter), and parameters $y_{\text{bottom}}$ (proliferation parameter value for $x \rightarrow 0$), $y_{\text{top}}$ (proliferation parameter value for $x \rightarrow \infty$), $n$ (Hill coefficient), and $EC50$ (half-maximal effective concentration).

**Limitations to proliferation data.** Variation in the plate reader-based growth measurements was observed. The magnitude of the read noise that could not be eliminated by postprocessing varied by channel ($A_{600}$, CFP, and YFP), as is exemplified in the data shown in Fig. 2C. Consequently, parameter estimates based on the noisy growth curve measurements have a limited technical precision. For growth rates, we evaluated the technical precision by comparing growth rate estimates of monocultures across different channels and for different parameter estimation ranges. Growth rates were generally found to be reproducible within approximately $\pm 0.025$ h$^{-1}$. For lag times and growth yields, we verified the correctness of the estimates by plotting the estimates on top of the growth curves. Manual visual inspection confirmed good agreement with the time points at which proliferation started and the final growth yields. We did not subject the proliferation estimates to any statistical analysis because of the presence of nonnegligible technical variation, which substantially contributed to the variation in the data.

**Evaluation of proliferation-dilution cycles.** Flow cytometry data were analyzed in custom Python (58) (version 3.8) scripts using the FlowCal (61) (version 1.3.0) package. Briefly, data were loaded and transformed into arbitrary units via FlowCal. Gating was directly performed in Python. The events were classified as YFP-positive–CFP-negative if FITC $> 1 \times 10^3$ a.u. and AmCyan $< 4 \times 10^3$ a.u., as YFP-negative–CFP-positive if FITC $< 7 \times 10^2$ a.u. and AmCyan $> 2 \times 10^3$ a.u., as YFP-negative–CFP-negative if FITC $< 1 \times 10^3$ a.u. and AmCyan $< 2 \times 10^3$ a.u., and as YFP-positive-CFP-positive otherwise. Some gating conditions were altered for transfer zero as cells precultured in the presence of methionine and harvested in midexponential phase showed slightly different fluorescence properties than cells cultured in the presence of Cbi and DMB and harvested in the stationary phase. In particular, the events were YFP-negative–CFP-positive if FITC $< 7 \times 10^2$ a.u. and AmCyan $> 5 \times 10^3$ a.u. and YFP-negative–CFP-negative if FITC $< 1 \times 10^3$ a.u. and AmCyan $< 5 \times 10^3$ a.u. All other gating was as described above. Population ratios (Sal:Dep) were determined as the ratio of the number of YFP-negative–CFP-positive events over the number of YFP-positive–CFP-negative events. The dynamic range of the population ratio estimation was determined in a transfer-specific manner by the ratio of the number of YFP-negative–CFP-positive events obtained when measuring a blank (or 1, if no such events where detected) over the total number of events collected per mixed cultures (i.e., $10^5$) and the ratio of the total number of events collected per mixed cultures (i.e., $10^5$) over the number of YFP-positive–CFP-negative events obtained when measuring a blank (or 1, if no such events where detected), respectively. For a discussion of YFP-negative–CFP-negative events and YFP-positive-CFP-positive events see Fig. S5.

**Corrinoid extraction and analysis.** Corrinoid extraction, cyanation, and purification were performed as previously described (44, 62). Briefly, 500 mL of cultures were harvested by centrifugation and corrinoids were extracted from the pellet with methanol and cyanated with 20 mg potassium cyanide/g (wet weight) of cells. Following cyanation, the samples were desalted using C18 SepPak (Waters Associates, Milford, MA). HPLC purification and profiling were performed as previously described (44).

**Statistical analysis.** Errors of EC$_{50}$ estimates are reported as 95% confidence intervals based on the t-statistic (degrees of freedom: number of data points minus 4) and the standard error of the estimate. Differences in EC$_{50}$ estimates extracted from four-parameter dose-response curves were statistically evaluated with two-tailed $t$ tests (H$_0$, identical EC$_{50}$; H$_1$, nonidentical EC$_{50}$; degrees of freedom, sum of number of data points in both samples minus 8). Differences in the estimated slopes of log$_{10}$(population ratio) were evaluated for each initial population ratio separately. For each initial ratio group, two-tailed $t$ tests (H$_0$, identical slopes; H$_1$, nonidentical slopes; degrees of freedom: sum of number of data points in both samples minus 4) were conducted for all 66 possible pairs of slopes. The global level of significance ($\alpha = 0.05$) was adjusted with the Holm-Bonferroni method to account for multiple comparisons. All statistical analysis was implemented in Python (58) (version 3.8).

**Materials availability.** All of the materials generated in this study will be made available on request, but we may require a completed material transfer agreement if there is potential for commercial application.

**Data and code availability.** All data and codes are available from the corresponding author upon reasonable request.

## SUPPLEMENTAL MATERIAL

Supplemental material is available online only.
**FIG S1**, TIF file, 0.5 MB.
**FIG S2**, TIF file, 0.3 MB.
**FIG S3**, TIF file, 1.7 MB.
**FIG S4**, TIF file, 0.5 MB.
**FIG S5**, TIF file, 1.9 MB.

**TABLE S1**, DOCX file, 0.02 MB.
**TABLE S2**, DOCX file, 0.01 MB.
**TABLE S3**, DOCX file, 0.01 MB.

## ACKNOWLEDGMENTS

This work was funded by National Institutes of Health grants R01GM114535 and R35GM139633 (M.E.T.).

We thank O. Sokolovskaya for initially suggesting the use of *E. coli* mutant strains. We thank K. Mok and O. Sokolovskaya for assistance with HPLC measurements and H. Nolla at the Flow Cytometry Facility in the Cancer Research Laboratory at UC Berkeley for assistance with flow cytometry measurements. We thank members of the Taga lab for helpful discussions. We also thank K. Kennedy, K. Mok, and Z. Hallberg for critical reading of the manuscript. *E. coli* strain MG1655 GB-1 was a kind gift from B.O. Palsson.

S.G. and M.E.T. conceived the research. S.G., G.J.P., and M.E.T. developed the *E. coli*-cobamide model system. S.G. designed and performed all experiments except corrinoid extraction and analysis. G.J.P. designed and performed corrinoid extraction and analysis. S.G. analyzed all data. S.G. and M.E.T. interpreted the data and wrote the manuscript with input from G.J.P.

We declare no conflict of interest.

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
