## [Reviewer comments · mSystems]

A salvaging strategy enables stable metabolite provisioning among free-living bacteria

Sebastian Gude, Gordon Pherribo, and Michiko Taga

Corresponding Author(s): Michiko Taga, University of California, Berkeley

Review Timeline:

Submission Date:	March 24, 2022
Editorial Decision:	April 25, 2022
Revision Received:	July 7, 2022
Accepted:	July 12, 2022

Editor: Alejandra Rodríguez-Verdugo

Reviewer(s): The reviewers have opted to remain anonymous.

Transaction Report:

DOI: <https://doi.org/10.1128/msystems.00288-22>

April 25, 2022

Dr. Michiko E Taga
University of California, Berkeley
Plant & Microbial Biology
111 Koshland Hall
Berkeley, CA 94720-3102

Re: mSystems00288-22 (A salvaging strategy enables stable metabolite provisioning among free-living bacteria)

Dear Dr. Michiko E Taga:

Thank you for submitting your manuscript to mSystems. We have completed our review and I am pleased to inform you that, in principle, we expect to accept it for publication in mSystems. However, acceptance will not be final until you have adequately addressed the reviewer comments.

The reviewers agree this study addresses an important question on microbial cross-feeding interactions. All three reviewers have made excellent suggestions for improvement that should be addressed before we can accept your work for publication. One of the main concerns was the overall lack of statistical analyses. Please provide the statistical tests used to interpret the data and figures. If possible, additional experiments should be performed to increase the sample size ($n=3$ being a low sample size). If this is not possible, please be explicit about the limitations of your data. In addition, two of the reviewers provided comments on how to further improve the discussion section. Overall, in your revision, you should ensure that the results include proper statistical analyses and that your discussion addresses both the limitations and generality of your findings. I look forward to receiving a revised manuscript.

Preparing Revision Guidelines

Sincerely,

Alejandra Rodríguez-Verdugo

Editor, mSystems

Journals Department
Reviewer comments:

Reviewer #1 (Comments for the Author):

The manuscript by Guide and coauthors addresses an often ignored mechanism of metabolite cross-feeding in microbial colleagues: the process of salvaging nutrients. This manuscript is timely and fills not only an important knowledge gap but also uses a nice experimental system to show that cells can employ precursor salvaging in order to alleviate metabolic dependencies in microbial communities. The experiments are well-designed and support the conclusions. That said, I believe the manuscript can still be improved with regard to the following points:

1. The use of `non-productive` populations can be misleading. Perhaps the authors should rephrase this to `non-producing members` throughout the manuscript.
2. The authors suggest (for instance in line 101) that precursor salvaging can be generalisable. However, the authors provide evidence for vitamin, especially cobamide biosynthesis. Perhaps the authors should discuss or mention how their findings can be generalised for other vitamin classes or metabolites. For instance, are other classes of metabolic pathways known to be incomplete amongst community members.
3. The authors should mention the appropriate statistical tests where required in the figures or the figure legends.
4. Line 146 and throughout the manuscript: Suggestion to change `jointly cultured` to `co-cultured`
5. Do the authors know or can speculate why the salvagers have growth benefits and smaller lag times than the dependents?

Reviewer #2 (Comments for the Author):

Please see the document attached to this email.

Reviewer #3 (Comments for the Author):

In the spirit of a full disclosure, I would like to reveal that I reviewed this article already before, when it has been submitted to a different journal before. I liked the work back then, and like it even better now. The authors have incorporated many of the suggestions I made to the previous version of this manuscript. As a consequence, its quality and overall readability has significantly improved. In general, I think that this manuscript addresses an important and open question in the field and provides a convincing answer. The work is well thought through and carefully done. The results are novel and shed new light on this type of metabolic interaction. However, below I list few points that should be addressed before the manuscript can be accepted for publication.

- (1) The authors argue that despite a lack of positive assortment, the interaction is stabilized because of a partial privatization of the public good. However, the authors neither show directly that the public good is partially privatized, nor is it necessary to explain the observed results. Unidirectional cross-feeding interactions cannot be exploited. So there is no Darwinian dilemma and no need to evoke a stabilizing mechanism. It will always be stable unless the dependent partner grows too slowly and dies out. This is an important issue. The presentation of the data should be adapted to more accurately represent this.
- (2) The manuscript is devoid of any statistics. This is probably, because the authors have only analyzed three replicates in each case. This is problematic, because it remains unclear whether or not the pattern is robust or just the consequence of a sampling error. A larger sample than 3 is generally recommended to allow to draw more general conclusions. Also, data shown in Figs. 2A, 3A-C, 4, and 5 should be analyzed statistically to support the authors' conclusions with an adequate test result.
- (3) I would advise to not only base the comparison of growth rates in the context of potential costs of metabolite production (e.g. line 187) on a visual inspection of the respective graphs, but to subject the corresponding data to a formal statistical analysis.

Here, the previously determined maximal growth rates could be statistically compared. Even better, however, would it be to determine the growth rates in mono- and coculture by plating and then compare these results. The optical density/ fluorescence measurements do not have the dynamic range necessary to distinguish differences in relative fitness. Therefore, the conclusion that the analysis of absolute fitness differences might be more important (lines 289-290), may not hold when the data is analyzed in a more sensitive way (i.e. by plating).

(4) The discussion starts already at the end of the results section (page 13). This is unusual and not necessary. I would recommend to move these parts entirely in the discussion section below. In general, I found the discussion to be rather short. I think it would be nice to expand this part a bit on different issues that arose during the experiment and the implications the gathered results might have in this and other systems.

Minor comments:

- lines 21, 57, 69, 99, 245: replace „non-productive" with „non-producing"
- line 75: consider replacing "processed" with "degraded"
- Line 254: remove "was"

Responses to reviewers:

Reviewer #1 (Comments for the Author):

The manuscript by Guide and coauthors addresses an often ignored mechanism of metabolite cross-feeding in microbial colleagues: the process of salvaging nutrients. This manuscript is timely and fills not only an important knowledge gap but also uses a nice experimental system to show that cells can employ precursor salvaging in order to alleviate metabolic dependencies in microbial communities. The experiments are well-designed and support the conclusions. That said, I believe the manuscript can still be improved with regard to the following points:

1. The use of `non-productive` populations can be misleading. Perhaps the authors should rephrase this to `non-producing members` throughout the manuscript.

R1.1:

We changed the wording as suggested.

2. The authors suggest (for instance in line 101) that precursor salvaging can be generalisable. However, the authors provide evidence for vitamin, especially cobamide biosynthesis. Perhaps the authors should discuss or mention how their findings can be generalised for other vitamin classes or metabolites. For instance, are other classes of metabolic pathways known to be incomplete amongst community members.

R1.2:

We added examples of potential precursor salvaging in the Discussion section (lines 368 – 370).

3. The authors should mention the appropriate statistical tests where required in the figures or the figure legends.

R1.3:

We now mention the appropriate statistical tests throughout the manuscript.

In particular, we provide confidence estimates for the extracted EC_{50} values and conducted two-tailed t-test whenever we compare EC_{50} values between conditions (Fig. 2 and Fig. 3).

Additionally, we included a statistical analysis of the population ratio time series data shown in Fig. 5C. The comparison of the differences in the rate of change in population ratio between co-cultures was assessed by two-tailed t-tests. Test results are stated in the main text where comparisons are made (lines 285 - 301).

For growth analyses, the plate reader-based growth curves were inherently noisy and the amplitude of the read noise varied by channel (as exemplified in Fig. 2C). The presence of these systematic, technical sources of variation in the data was non-negligible, thus falsifying a key assumption of statistical analysis (i.e., that variation is mainly caused by statistical factors such as sampling errors). Therefore, in order to communicate the error in these measurements, we report their technical precision in the figure legends (Fig. 2, Fig. 3, Fig. 4, and Fig. 5), in the main text (lines 133 and 191 - 192), and in the Methods (lines 652 - 665).

4. Line 146 and throughout the manuscript: Suggestion to change `jointly cultured` to `co-cultured`

R1.4:

We changed the wording as suggested.

5. Do the authors know or can speculate why the salvagers have growth benefits and smaller lag times than the dependents?

R1.5:

Growth benefits and smaller lag times of salvagers relative to dependents suggest that salvagers can maintain preferential access to salvaged cobamides. It could be that salvagers sequester some portion of the salvaged cobamides for private use prior to releasing them into the extracellular environment. We now explicitly mention these points in the Discussion section (lines 335 – 343).

Reviewer #2 (Comments for the Author):

Gude et al nicely demonstrate that partial privatization can protect producers of B12 from competitive exclusion by bacteria that consume but do not produce B12. The authors first demonstrate that an E. coli genotype that is dependent on external B12, can acquire B12 from a salvaging strain of E. coli that produces B12 from the precursor Cbi. The authors then show that the growth rate of both strains in co-culture is influenced by the type of B12 lower ligand that is provided in the media. It is further shown that Cbi can inhibit the growth of E. coli at intermediate Cbi/B12 ratios. The authors finally demonstrate that salvagers cannot be outcompeted by dependents even if the dependents have faster maximum growth rate. The authors argue that partial privatization provides an explanation for the observation of B12 salvaging and dependent strategies in natural populations, and more broadly for the maintenance of metabolite provisioning generally in microbial communities.

This work elegantly demonstrates that partial privatization can allow B12 salvagers to avoid being outcompeted by B12 dependent strains. This is in line with previous demonstrations in other microbial systems that partial privatization generates snow drift games that can maintain producers of public goods. The question that the work does not fully address however, is why complete privatization is not observed. In theory a strain that only salvaged as much B12 as it needed and did not release B12 would be more fit than the salvaging strain used here that produces more B12 than needed. The authors argue that there is no cost of salvaging because their salvaging strain grows as quickly on B12 as it does on Cbi+DMB. However, the strain used here appears to constitutively express the salvaging pathway suggesting that it pays costs of salvaging even when it is not actively engaged in salvaging. A strain that privatized B12 production by regulating its salvaging could in theory avoid the cost of overexpression. The explanation of why complete privatization is not observed may in part be explained by the observation that Cbi can inhibit growth of the salvager. Explicitly including some discussion of a complete privatizing strategy would further strengthen the manuscript.

R2.1:

We thank the reviewer for this suggestion. We expanded the Discussion section based on the reviewer's comments (lines 323 - 330).

Minor comments:

Line 183 – I was not entirely clear why the salvagers were able to maintain higher growth rates at low concentrations of 2MA than DMB. If the cell can generate internal ligand shouldn't this allow for similar growth as either lower ligand reduces in concentration?

R2.2:

Yes, indeed, similar growth would be expected for low concentrations of the two lower ligands. The reviewer rightly observed that salvager's growth rates at low lower ligand concentrations (Fig. 3A (lower ligand: DMB) and Fig. 3B (lower ligand: 2MA)) are not similar. We account this discrepancy to experiment-to-experiment variation in the level of internally produced lower ligand. Consistent with this explanation, we find that the growth rates of salvagers in the absence of lower ligand supplementation are in good agreement with the growth rates observed for the lowest measured level of lower ligand supplementation (Fig. 3A: 0.436 1/h [0 M DMB] and 0.439 1/h [100 fM DMB]; Fig. 3B: 0.269 1/h [0 M DMB] and 0.251 1/h [1 pm 2MA]). We now mention this fact explicitly in the legend of Fig. 3.

Line 254 – There is an extra “was”

R2.3:

We fixed the typo.

Line 259 – Was the carrying capacity set by glycerol or B12 precursors? It seems one might expect different dynamics if competition is primarily for B12 or carbon.

R2.4:

Competition was primarily for glycerol as the B12 precursors were supplied in excess. We now specify this fact in the text (lines 263 - 265).

Reviewer #3 (Comments for the Author):

In the spirit of a full disclosure, I would like to reveal that I reviewed this article already before, when it has been submitted to a different journal before. I liked the work back then, and like it even better now. The authors have incorporated many of the suggestions I made to the previous version of this manuscript. As a consequence, its quality and overall readability has significantly improved. In general, I think that this manuscript addresses an important and open question in the field and provides a convincing answer. The work is well thought through and carefully done. The results are novel and shed new light on this type of metabolic interaction. However, below I list few points that should be addressed before the manuscript can be accepted for publication.

R3.0:

We thank the reviewer for taking the time to review our work twice.

(1) The authors argue that despite a lack of positive assortment, the interaction is stabilized because of a partial privatization of the public good. However, the authors neither show directly that the public good is partially privatized, nor is it necessary to explain the observed results. Unidirectional cross-feeding interactions cannot be exploited. So there is no Darwinian dilemma and no need to evoke a stabilizing mechanism. It will always be stable unless the dependent partner grows too slowly and dies out. This is an important issue. The presentation of the data should be adapted to more accurately represent this.

R3.1:

Firstly, we agree with the reviewer that we did not directly show that the salvaged cobamides are partially privatized. Such a direct demonstration would have been elegant. Unfortunately, the molecular basis of cobamide release in *E. coli* is unknown to date, making a direct study of cobamide release challenging. We would like to point out that we present indirect evidence for

partial privatization. Namely, the observation of an extended lag phase experienced by dependents when co-cultured with salvagers (Fig. 2C). This finding is consistent with a cobamide release mode in which salvagers sequester a portion of the salvaged cobamides inside the cell for private use prior to releasing them into the extracellular environment as a 'public good'. We revised the manuscript to account for the lack of direct proof of partial privatization and consequently only mention privatization as a speculation in the Discussion section (lines 312, 317, 325, and 338).

Secondly, we disagree with the reviewer that a unidirectional cross-feeding interaction cannot be exploited. If the producing species releases metabolites into the environment too generously while not maintaining private access to (or not re-internalizing) a portion of it, the producing species can make itself temporarily dispensable. This scenario can provide an opportunity for the non-producing species to overexploit the producing species and to arrest its proliferation by quickly consuming the publicly available metabolites (e.g., if it has a much faster growth rate). If such a proliferation arrest persists sufficiently long, the productive species may get lost (e.g., diluted out). The loss of the productive species would then ultimately lead to a collapse of the entire metabolically interconnected population. As it was previously unclear whether and to what extent salvaged cobamides are released from salvagers, the above-mentioned scenario could have been possible. We revised the manuscript in an attempt to more clearly communicate how we envision overexploitation could have happened (lines 247 - 254).

(2) The manuscript is devoid of any statistics. This is probably, because the authors have only analyzed three replicates in each case. This is problematic, because it remains unclear whether or not the pattern is robust or just the consequence of a sampling error. A larger sample than 3 is generally recommended to allow to draw more general conclusions. Also, data shown in Figs. 2A, 3A-C, 4, and 5 should be analyzed statistically to support the authors' conclusions with an adequate test result.

R3.2 (same as R1.3):

We now mention the appropriate statistical tests throughout the manuscript.

In particular, we provide confidence estimates for the extracted EC_{50} values and conducted two-tailed t-test whenever we compare EC_{50} values between conditions (Fig. 2 and Fig. 3).

Additionally, we included a statistical analysis of the population ratio time series data shown in Fig. 5C. The comparison of the differences in the rate of change in population ratio between co-cultures was assessed by two-tailed t-tests. Test results are stated in the main text where comparisons are made (lines 285 - 301).

For growth analyses, the plate reader-based growth curves were inherently noisy and the amplitude of the read noise varied by channel (as exemplified in Fig. 2C). The presence of these systematic, technical sources of variation in the data was non-negligible, thus falsifying a key assumption of statistical analysis (i.e., that variation is mainly caused by statistical factors such as sampling errors). Therefore, in order to communicate the error in these measurements, we report their technical precision in the figure legends (Fig. 2, Fig. 3, Fig. 4, and Fig. 5), in the main text (lines 133 and 191 - 192), and in the Methods (lines 652 - 665).

(3) I would advise to not only base the comparison of growth rates in the context of potential costs of metabolite production (e.g. line 187) on a visual inspection of the respective graphs, but to subject the corresponding data to a formal statistical analysis. Here, the previously determined maximal growth rates could be statistically compared. Even better, however, would it be to determine the growth rates in mono- and coculture by plating and then compare these results. The optical density/ fluorescence measurements do not have the dynamic range necessary to distinguish differences in relative fitness. Therefore, the conclusion that the analysis of absolute fitness differences might be more important (lines 289-290), may not hold

when the data is analyzed in a more sensitive way (i.e. by plating).

R3.3:

We agree with the reviewer's assessment of the technical limitations of plate reader measurements. We see now that we hadn't communicated these limitations, and how they may impact our conclusions, clearly in the manuscript. We revised the manuscript accordingly. Particularly, we now describe how we assessed the technical precision of the growth measurements (Methods – "Limitations to proliferation data", lines 652 - 665), state the technical precision in the figure legends (Fig. 2, Fig. 3, Fig. 4, and Fig. 5), and explicitly point out the technical precision when necessary in the main text (lines 133 and 191 - 192).

Though the plate reader measurements have technical limitations, we would like to argue that they are sufficient to make the following points:

- 1) To distinguish in which conditions strains can proliferate and in which not (Fig. 2B, Fig. 3A-C, Fig. 4 ABD, and Fig. 5A).
- 2) That GB-1 mutants have a much higher growth rate compared to WT strains (Fig. 2B and Fig. 5A).
- 3) The general direction of changes in growth yield (Fig. 5B).
- 4) The presence of drastic lag times (Fig. 2C and Fig. 4B).

Yet, the plate reader data is indeed not sufficiently precise to make more nuanced points, e.g., whether the growth rates of salvager and dependent pairs are identical or not as the detected differences are comparable to the estimated technical precision of the growth rate (see Methods – "Limitations to proliferation data" for details). We revised the text to clearly communicate this limitation and adjusted our conclusions accordingly. Namely, we explicitly mention the technical precision of growth rate estimation whenever growth rates differences are small and reworded our conclusion about a potential metabolic burden caused by salvaging accordingly (lines 132 - 133, 190 - 193, and 299 - 301). Furthermore, in the Discussion section, we now explicitly mention the possibility that undetected growth rate differences may explain the observed co-culture dynamics before speculating about other potential explanations (lines 348 - 350). Our main goal was not to prove that there is no growth rate difference between the strains but rather to highlight potential alternative causes for our observation. The previous characterization of the GB-1 mutant by others (Herring et al, 2006; Cheng et al, 2014) provided a rare opportunity for such an endeavor as such in-depth information is rarely available for (mutant) strains used in the context of microbial ecology.

Due to the non-negligible presence of systematic, technical variations in the growth data, we did not conduct statistical analyses on the growth data as explained above.

(4) The discussion starts already at the end of the results section (page 13). This is unusual and not necessary. I would recommend to move these parts entirely in the discussion section below. In general, I found the discussion to be rather short. I think it would be nice to expand this part a bit on different issues that arose during the experiment and the implications the gathered results might have in this and other systems.

R3.4:

We agree with the reviewer's suggestion and moved the mentioned text to the Discussion section which was overall reworked and expanded (lines 346 - 359).

Minor comments:

- lines 21, 57, 69, 99, 245: replace „non-productive" with „non-producing"

R3.5:

We changed the wording as suggested throughout the manuscript.

- line 75: consider replacing "processed" with "degraded"

R3.6:

We decided to retain the original wording.

- Line 254: remove "was"

R3.7:

We fixed the typo.

July 12, 2022

Dr. Michiko E Taga
University of California, Berkeley
Plant & Microbial Biology
111 Koshland Hall
Berkeley, CA 94720-3102

Re: mSystems00288-22R1 (A salvaging strategy enables stable metabolite provisioning among free-living bacteria)

Dear Dr. Michiko E Taga:

Thank you for sending your revised manuscript and for your detailed responses to the reviewers' comments. The manuscript has improved and it's now ready for publication. Thank you for the privilege of reviewing this exciting study.

Your manuscript has been accepted, and I am forwarding it to the ASM Journals Department for publication. For your reference, ASM Journals' address is given below. Before it can be scheduled for publication, your manuscript will be checked by the mSystems production staff to make sure that all elements meet the technical requirements for publication. They will contact you if anything needs to be revised before copyediting and production can begin. Otherwise, you will be notified when your proofs are ready to be viewed.

Publication Fees:

If you would like to submit a potential Featured Image, please email a file and a short legend to mSystems@asmusa.org. Please note that we can only consider images that (i) the authors created or own and (ii) have not been previously published. By submitting, you agree that the image can be used under the same terms as the published article. File requirements: square dimensions (4" x 4"), 300 dpi resolution, RGB colorspace, TIF file format.

We recognize that the video files can become quite large, and so to avoid quality loss ASM suggests sending the video file via <https://www.wetransfer.com/>. When you have a final version of the video and the still ready to share, please send it to mSystems staff at mSystems@asmusa.org.

Sincerely,

Alejandra Rodríguez-Verdugo
Editor, mSystems

Journals Department
Supp. Fig. 3: Accept
Supp. Fig. 4: Accept
Supp. Table 3: Accept
Supp. Table 2: Accept
Supp. Fig. 5: Accept
Supp. Table 1: Accept
Supp. Fig. 1: Accept
Supp. Fig. 2: Accept